# F²ED-LEARNING: GOOD FENCES MAKE GOOD NEIGHBORS

## ABSTRACT

In this paper, we present F²ED-LEARNING, the first federated learning protocol simultaneously defending against both a semi-honest server and Byzantine malicious clients. Using a robust mean estimator called FilterL2, F²ED-LEARNING is the first FL protocol providing dimension-free estimation error against Byzantine malicious clients. Besides, F²ED-LEARNING leverages secure aggregation to protect the clients from a semi-honest server who wants to infer the clients' information from the legitimate updates. The main challenge stems from the incompatibility between FilterL2 and secure aggregation. Specifically, to run FilterL2, the server needs to access individual updates from clients while secure aggregation hides those updates from it. We propose to split the clients into shards, securely aggregate each shard's updates and run FilterL2 on the updates from different shards. The evaluation shows that F²ED-LEARNING consistently achieves optimal or close-to-optimal performance under three attacks among five robust FL protocols.

## 1 INTRODUCTION

Federated learning (FL) has drawn numerous attention in the past few years as a new distributed learning paradigm. In federated learning, the users collaboratively train a model with the help of a centralized server when all the data is held locally to preserve the users' privacy. The privacy guarantee can be further enhanced using secure aggregation technique (Bonawitz et al., 2017) which hides the individual local updates and only reveals the aggregated global update. The graceful balance between utility and privacy popularizes federated learning in a variety of sensitive applications such as Google GBoard, healthcare service and self-driving cars.

The above threat model assumes that all the users honestly upload their local updates. However, it is likely that a small number of clients are malicious in a large-scale FL system with tens of thousands of clients. Besides, in most SGD-based FL algorithms used today (McMahan & Ramage, 2017), the centralized server averages the local updates to obtain the global update, which is vulnerable to even only one malicious client. Therefore, a malicious client can arbitrarily craft its update to either prevent the global model from converging or lead it to a sub-optimal minimum. This kind of attack in federated learning is well-studied by Bhagoji et al. (2019); Fang et al. (2019); Bagdasaryan et al. (2020); Sun et al. (2020).

To mitigate these attacks, various Byzantine-robust FL protocols (Blanchard et al., 2017; Yin et al., 2018; Fu et al., 2019; Pillutla et al., 2019) are proposed to reduce the impact of the contaminated updates. These protocols replace trivial averaging with well-designed Byzantine-robust mean estimators. These estimators suppress the influence of the malicious updates and output a mean estimation as accurate as possible. Nevertheless, almost all of these aggregators suffer from the curse of dimensionality. Specifically, the estimation error scales up with the size of the model in a square-root fashion. As a concrete example, a three-layer MLP on MNIST contains more than 50,000 parameters and leads to a 223-fold increase of the estimation error, which is prohibitive in practice. Draco (Chen et al., 2018), BULYAN (Mhamdi et al., 2018) and ByzantineSGD (Alistarh et al., 2018) are the only three works that state to yield dimension-free estimation error. However, Draco is designed for distributed learning and is incompatible with federated learning because it requires redundant updates from each worker. On the other hand, although Bulyan (Mhamdi et al., 2018) and ByzantineSGD (Alistarh et al., 2018) provide dimension-free estimation error, it is based on

much stronger assumptions than other works. When the assumptions are relaxed to the common case, Bulyan's estimation error still scales up with the square root of the model size as discussed in Section 2.

In addition, these robust FL protocols have incompatible implementation with secure aggregation techniques. The robust estimators have to access local updates while secure aggregation hides them from the server. Consequently, the system cannot simultaneously protect the server and the clients, but has to place complete trust in either of them. The lack of two-way protection severely harms the people's confidence in the FL system and prevents federated learning from being used in many sensitive applications such as home monitoring and self-driving cars.

**Contribution.** In this paper, we propose FEDERATED LEARNING WITH FENCE, abbreviately $F^2$ED-LEARNING. $F^2$ED-LEARNING integrates a robust mean estimator with dimension-free error (Steinhardt, 2018) and secure aggregation (Bonawitz et al., 2017) to defend against both the Byzantine malicious clients and the semi-honest server. In particular, $F^2$ED-LEARNING is the first Byzantine-robust FL system with dimension-free estimation error. To address the incompatibility, the clients are split into multiple shards, the local updates from the same shard are securely aggregated at the centralized server, and the robust estimator is run on the aggregated local updates from different shards. Surprisingly, sharding also consolidates the independently and identically distributed (IID) assumption required by the robust estimator even under heterogeneous data distribution. According to Lindeberg central limitation theorem (Lindeberg, 1922), despite the heterogeneity of the individual local updates, the aggregated local updates from the shards will approximately follow an IID Gaussian distribution.

## 2    LOOPHOLE IN BULYAN AND BYZANTINESGD & RELATED WORK

Byzantine-robust aggregation has drawn enormous attention in the past few years due to the emergence of various distributed attacks in federated learning. Fang et al. (2019) formalize the attack as an optimization problem and successfully migrate the data poisoning attack to federated learning. The proposed attacks even work under Byzantine-robust federated learning. Sun et al. (2020) manage to launch data poisoning attack on the multi-task federated learning framework. Bhagoji et al. (2019) and Bagdasaryan et al. (2020) even manage to insert backdoor functionalities into the model via local model poisoning or local model replacement.

A variety of Byzantine-robust FL protocols are proposed to defend against these attacks. Krum (Blanchard et al., 2017) picks the subset of updates with enough close neighbors and averages the subset. Yin et al. (2018) leverage traditional robust estimators like trimmed mean or median to achieve order-optimal statistical error rate under strongly convex assumptions. Yin et al. (2019) propose to use robust mean estimators to defend against saddle point attack. Mhamdi et al. (2018) pointed out that Krum, trimmed mean and median all suffers from $\mathcal{O}(\sqrt{d})$ ($d$ is the model size) estimation error and proposed a general framework Bulyan to reduce the error to $\mathcal{O}(1)$. However, we point out that the improvement of Bulyan actually comes from its stronger assumption. In particular, Bulyan assumes that expectation of the distance between two benign updates is bounded by a constant $\sigma_1$, while Krum assumes that the distance is bounded by $\sigma_2\sqrt{d}$. We can easily see that if $\sigma_1 = \sigma_2\sqrt{d}$, Bulyan falls back to the same order of estimation error as Krum. The same loophole exists in the analysis of ByzantineSGD (Alistarh et al., 2018). Consequently, there is no known federated learning protocol with dimension-free estimation error against Byzantine adversaries.

## 3    PROBLEM SETUP

In this section, we review the general pipeline of federated learning, introduce the threat model, and establish the notation system. We use bold lower-case letters (*e.g.* **a**,**b**,**c**) to denote vectors, and bold upper-case letters (*e.g.* **A**, **B**, **C**) for matrices. We denote $1 \cdots n$ with $[n]$.

**Federated Learning Pipeline.**    In a federated learning system, there are one server $\mathcal{S}$ and $n$ clients $\mathcal{C}_i, i \in [n]$. Each client holds data samples drawn from some unknown distribution $\mathcal{D}$. Let $\ell(\mathbf{w}; \mathbf{z})$ be the loss function on the model parameter $\mathbf{w} \in \mathbb{R}^d$ and a data sample $\mathbf{z}$. Let $\mathcal{L}(\mathbf{w}) = \mathbb{E}_{\mathbf{z} \sim \mathcal{D}}[\ell(\mathbf{w}; \mathbf{z})]$ be the population loss function. Our goal is to learn the model $\mathbf{w}$ such that the population loss function

is minimized:

$$\mathbf{w}^* = \arg \min_{\mathbf{w} \in \mathcal{W}} \mathcal{L}(\mathbf{w}).$$

To learn $\mathbf{w}^*$, the whole system runs a $T$-round federated learning protocol. Initially, the server stores a global model $\mathbf{w}_0$. In the $t^{th}$ round, $\mathcal{S}$ broadcasts the global model $\mathbf{w}_{t-1}$ to the $m$ clients. The clients then run the local optimizers (*e.g.* SGD, Adam, and RMSprop), compute the difference $\mathbf{g}_t^{(i)}$ between the optimized model and the global model, and upload the difference to $\mathcal{S}$. In the $t^{th}$ round, $\mathcal{S}$ takes the average of the differences and update the global model $\mathbf{w}_t = \mathbf{w}_{t-1} + \frac{1}{n} \sum_{i=1}^{n} \mathbf{g}_t^{(i)}$.

**Threat Model & Defence Goal.**  We assume that the centralized server $\mathcal{S}$ is semi-honest. The server can launch whatever attacks such as inference attack using legitimate updates from the clients as the only inputs. However, the server cannot deviate from the protocol for the sake of regulation or reputation pressure. Specifically, we want to emphasize there is no collusion between the server and the clients. On the other hand, we assume that the clients are $\epsilon$-Byzantine malicious, which means at most $\epsilon m$ clients can be malicious. Malicious clients can arbitrarily deviate from the protocol and tamper with their own updates without being detected.

In this paper, we aim to achieve a dimension-free error for the mean estimation in each round. Let $\boldsymbol{\mu}$ be the true mean of the benign distribution and the output of a protocol with contaminated inputs be $\hat{\boldsymbol{\mu}}$. The estimation error is defined by the $\ell_2$ distance between the true mean and the estimation $\|\hat{\boldsymbol{\mu}} - \boldsymbol{\mu}\|_2$.

## 4 F$^2$ED-LEARNING: ROBUST PRIVACY-PRESERVING DISTRIBUTED FL

In this section, we formally present our main protocol: F$^2$ED-LEARNING. We first introduce F$^2$ED-LEARNING step by step and formally establish the robustness and security guarantees. Then we discuss the effect of sharding on the IID distribution assumption.

### 4.1 F$^2$ED-LEARNING: BYZANTINE-ROBUST PRIVACY-PRESERVING FEDERATED LEARNING

The complete F$^2$ED-LEARNING protocol is presented in Algorithm 1. F$^2$ED-LEARNING iteratively executes the following steps: (1) the server broadcasts the global model to the clients; (2) clients train the global model with their local data; (3) clients in the same shard run secure aggregation protocol to upload the mean of their updates to the server; (4) the server aggregates the received updates using robust mean estimation; (5) the server updates the global model with the aggregated global update. We highlight step (3) and (4) newly proposed in F$^2$ED-LEARNING.

**Sharded Secure Aggregation (line 8-10, 12, 16).**  Secure aggregation is developed by Bonawitz et al. (2017) to defend against the honest but curious server in federated learning. Secure aggregation allows the server to obtain the sum of the clients' updates but hides the individual updates cryptographically. We introduce an oversimplified version of secure aggregation as follows for the ease of clarification. As the first step, each client samples random values for the other clients and send the values to the corresponding clients (line 8-10). After receiving all the values from other clients, each client sums up the received values and subtracts the values generated by itself to produce a random mask (line 12). Each client blinds its local update with the random mask and sends the blinded update to the server (line 13). The server then sums up all the blinded updates and obtains the summed update in plaintext (line 15). Obviously, all the masks cancel out during aggregation and the server receives the plaintext sum. Secure aggregation provides strong privacy guarantee for the clients that the server cannot see anything but the aggregated global update and each client is hidden in thousands of other clients.

However, in our threat model, vanilla secure aggregation is insufficient since it provides no protection for the server. As the individual updates are completely hidden, there is no way that the server can identify the malicious clients even after detecting the attack. To address the issue, we propose to split the clients into multiple shards and run secure aggregation within each shard. The size of the shards provides a trade-off between the protection for the server or the clients. The smaller the size is, the more information is revealed to the server, thus the easier to defend against Byzantine malicious clients and the harder to fight off the semi-honest server. The trade-off is discussed in detail in Section 4.2.

---

**Algorithm 1:** $F^2$ED-LEARNING: Robust Privacy-Preserving Sharded Federated Learning.

---

1 **for** $t \leftarrow [T]$ **do**
2    **Server:**
3       Split $n$ clients into $p$ shards $\{H_j\}_{j \in [p]}$
4       Broadcast $\{H_j\}_{j \in [p]}$ and the global model $\mathbf{w}_{t-1}$ to all the clients
5    **Client:**
6       **foreach** *client* $i \in [n]$ **do**
7          Locate its own shard $j$
8          Generate random masks $\mathbf{u}_{ik}^{(j)}, k \in H_j/i$
9          **foreach** $k \in H_j/i$ **do**
10             Send $\mathbf{u}_{ik}$ to $k$
11          Train the local model $\mathbf{w}_t^{(i)}$ using $\mathbf{w}_t$ as initialization
12          $\mathbf{g}_t^{(i)} = \mathbf{w}_t^{(i)} - \mathbf{w}_{t-1} + \sum_{k \neq i, i \in H_j, k \in H_j} \mathbf{u}_{ik}^{(j)} - \sum_{k \neq i, i \in H_j, k \in H_j} \mathbf{u}_{ki}^{(j)}$
13          Send $\mathbf{g}_t^{(i)}$ to the server
14    **Server:**
15       **foreach** $H_j \in \{H_j\}_{j \in [p]}$ **do**
16          $\mathbf{g}_t^{H_j} = \sum_{k \in H_j} \mathbf{g}_t^{(k)}$
17       $\mathbf{g}_t = \frac{1}{|H_j|} \text{FilterL2}(\{\mathbf{g}_t^{H_j}\}_{j \in [p]})$
18       $\mathbf{w}_t = \mathbf{w}_{t-1} + \mathbf{g}_t$

---

**Robust Mean Estimation (line 17).** The core step in Byzantine-robust federated learning is to estimate the true mean of the benign updates as accurate as possible even with some malicious clients. The most commonly used aggregator, averaging, is proven to be vulnerable to even only one malicious client. All other works addressing the issue such as Krum (Blanchard et al., 2017) and Bulyan (Mhamdi et al., 2018) suffer from a dimension-dependent estimation error. Such error is unacceptable even for training a 3-layer MLP on MNIST, not to mention more complicated tasks and models such as VGG16 or ResNet50.

Actually, the above problem is well studied in statistics under the name "robust mean estimation" and there already exist several robust mean estimators with dimension-free estimation error (Diakonikolas et al. (2019); Charikar et al. (2017); Steinhardt (2018); Cheng et al. (2019); Dong et al. (2019)). Therefore, instead of reinventing the wheel, we choose to leverage a representative robust mean estimator: FilterL2 (Algorithm 2). The following formulation is related to the presentation given in Steinhardt (2018).

---

**Algorithm 2:** FilterL2: dimension-free robust mean estimation (Steinhardt (2018)).

---

**Input:** $\mathbf{g}_1, \cdots, \mathbf{g}_n \in \mathbb{R}^d, \eta > 1$
1 Let $c_1, \cdots, c_n = 1$
2 $\hat{\boldsymbol{\mu}}_c = (\sum_{i=1}^n c_i \mathbf{g}_i) / (\sum_{i=1}^n c_i)$
3 $\hat{\boldsymbol{\Sigma}}_c = (\sum_{i=1}^n c_i (\mathbf{g}_i - \hat{\mu}_c)(\mathbf{g}_i - \hat{\mu}_c)^\top) / (\sum_{i=1}^n c_i)$
4 Let $\mathbf{v}$ be the maximum eigenvector of $\hat{\boldsymbol{\Sigma}}_c$, and let $\hat{\sigma}_c^2 = \mathbf{v}^\top \hat{\boldsymbol{\Sigma}}_c \mathbf{v}$
5 **if** $\hat{\sigma}_c^2 \leq \eta \sigma^2$ **then** return $\hat{\boldsymbol{\mu}}_c$
6 **else** let $\tau = \langle \mathbf{x}_i - \hat{\boldsymbol{\mu}}_c \rangle^2$, and update $c_i \leftarrow c_i \cdot (1 - \tau_i / \tau_{\max})$, where $\tau_{\max} = \max_i \tau_i$
7 Go back to line 2

---

Specifically, FilterL2 assigns each update a weight and iteratively updates the weights until the weights for the malicious updates are small enough. As mentioned, FilterL2 provides dimension-free error rate formally presented as follows.

**Theorem 1** (Steinhardt (2018)). *Let $D$ be the honest dataset and $D^*$ be the contaminated version of $D$ by inserting malicious samples. Suppose that $|D^*| \leq |D|/(1 - \epsilon), \epsilon \leq \frac{1}{12}$, and further*

*suppose that* $\text{MEAN}[D] = \boldsymbol{\mu}$ *and* $\|\text{COV}[D]\|_{op} \leq \sigma^2$. *Then given* $\mathcal{D}^*$, *Algorithm 2 outputs* $\hat{\boldsymbol{\mu}}$ *s.t.* $\|\hat{\boldsymbol{\mu}} - \boldsymbol{\mu}\|_2 = \mathcal{O}(\sigma\sqrt{\epsilon})$ *using* $\text{POLY}(n, d)$ *time.*

Although Algorithm 2 only takes polynomial time to run, the per-round time complexity is $\mathcal{O}(nd^2)$ if implemented with power iteration. Given $d$ is large, the running time is still quite expensive in practice. To address the issue, we cut the update vectors into $k$ sections and apply the robust estimator to each of the sections. The acceleration scheme reduces the per-round running time to $\mathcal{O}(nd^2/k)$ but increases the estimation error to $\mathcal{O}(\sigma\sqrt{k})$. For instance, if we take $k = \sqrt{d}$, the per-round running time becomes $\mathcal{O}(nd)$ while the estimation error grows to $\mathcal{O}(\sigma\sqrt[4]{\sigma^2 d})$. Despite the compromise for acceleration, FilterL2 still gives the known optimal estimation error and outperforms other robust FL protocols by multiple magnitudes.

## 4.2 ROBUSTNESS & SECURITY ANALYSIS

In this section, we rigorously present the security and robustness guarantee of $\text{F}^2\text{ED-LEARNING}$.

**Security Guarantee.** We first give the security guarantee of $\text{F}^2\text{ED-LEARNING}$ as follows. Intuitively, no more information about the clients except the averaged updates from the shards is revealed to the centralized server. Thus, each client's update is hidden in all the other clients in its shard. The proof is deferred to A.

**Corollary 1** (Security against honest-but-curious server; Informal). *There exists a PPT (probabilistic polynomial Turing machine) simulator which can only see the averaged updates from the shards and its output is computationally indistinguishable from the transcript of* $\text{F}^2\text{ED-LEARNING}$.

**Robustness Guarantee.** We now give the formal robustness guarantee of $\text{F}^2\text{ED-LEARNING}$. The proof is deferred to Appendix B.

**Corollary 2** (Robustness against Byzantine adversaries). *Given the number of clients $n$, the number of shards $p$ and the fraction of corrupted clients $\epsilon$,* $\text{F}^2\text{ED-LEARNING}$ *provides a mean estimation with dimension-free error as long as* $12\epsilon n < p$.

**Remark.** Given the formal security and robustness guarantee, we can see that $\text{F}^2\text{ED-LEARNING}$ actually provides a convenient way to calibrate the protection for the server or the clients. Concretely, $\text{F}^2\text{ED-LEARNING}$ can tolerate up to $\lfloor \frac{p}{12} \rfloor - 1$ malicious clients and hide each honest client's update in the mean of $\lfloor \frac{n}{p} \rfloor$ updates.

## 4.3 DISCUSSION ON THE I.I.D. DISTRIBUTION ASSUMPTION IN COROLLARY 2

To derive Corollary B, we assume that the updates from the benign clients are drawn IID from some distribution $\mathcal{D}$. In this section, we explore the rationality of the assumption.

**Source of Non-IID updates in FL.** It is well known that in FL, data is heterogeneously distributed across clients. Therefore, the collected updates typically do not follow IID distribution under any proper distribution. Another source of non-IID updates in FL is the random initialization of local models. As known, many neural networks is permutation-invariant. For instance, in a two-layer fully connected network, the neurons in the two layers can be permuted correspondingly without changing the functionality of the network. Therefore, even with the same training data, different initialization can lead to different models within the same permutation-invariant class.

To overcome the second issue, we take the old-fashioned solution by requiring the clients to share the same initialization before the training phase starts. Note that there is a line of work (Yurochkin et al., 2019a;b; Wang et al., 2020) focusing on addressing the issue using matching algorithm and Bayesian non-parametric model. We deem it as an interesting future direction to integrate these works in $\text{F}^2\text{ED-LEARNING}$.

For the rest of the section, we ignore the second issue and focus on the first issue. We formally model the heterogeneous data distribution under some explicit assumptions and discuss how sharding addresses the first issue under such assumptions. Note that with sharding we do not solve the slow convergence issue in FL due to non-IID updates. Instead, we only create an IID distribution among

shards to solidate the IID assumption required in the proof of 2. However, the distribution is highly biased and still suffers from low convergence rate due to the intrinsic non-IID data distribution.

**From non-I.I.D. to I.I.D.** It is a widely accepted assumption in traditional distributed learning theory that the updates should be independently and identically distributed. The assumption is reasonable in the sense that the server can decide how to distribute the data to the workers in traditional distributed learning. However, in federated learning, the data is generated by the clients locally so the updates are not necessarily and typically not IID distributed. This poses a challenge on the robustness analysis. Now we propose a novel perspective to conduct robustness analysis in federated learning. Succinctly, by aggregating the shards first, we are able to reduce the non-IID distribution to an IID distribution. As the first step, we model the heterogeneous update distribution in federated learning as follows.

**Definition 1** (Heterogeneous Distribution). *Let $\mathbb{D}$ be a set of $k$ distributions $\mathbb{D} = \{\mathcal{D}_i\}_{i \in [k]}$ where $\mathbb{E}[\mathcal{D}_i] = \mu_i$ and $\mathbb{V}[\mathcal{D}_i] = \sigma_i^2$. Each client $\mathcal{C}_j$'s update $\mathbf{g}_j$ follows a distribution $\mathcal{D}_{\phi(j)}$ where $\phi$ is a mapping from the client index to the distribution index.*

Note that in the definition we use scalar data for the ease of clarification. The formalization can be easily extended to data vectors by separately considering each dimension. The definition captures the most important feature that each client's update is drawn from different distributions.

As the second step, we analyze the influence of sharding on the update distribution. Surprisingly, sharding pushes the non-IID distribution to a well-regulated IID distribution according to Lindeberg central limit theorem.

**Theorem 2** (Lindeberg Central Limit Theorem (Linnik (1959))). *Suppose $\{X_1, \cdots, X_n\}$ is a sequence of independence random variables (not necessarily identically distributed), each with finite expected value $\mu_i$ and variance $\sigma_i^2$. Define $s_n^2 = \sum_{i=1}^n \sigma_i^2$. Suppose that $\forall \epsilon > 0$,*

$$\lim_{n \to \infty} \frac{1}{s_n^2} \sum_{i=1}^n \mathbb{E}[(X_i - \mu_i)^2 \cdot \mathbb{1}\{|X_i - \mu_i| > \epsilon s_n\}] = 0.$$

*Then the distribution of the standardized sums converges towards the standard normal distribution.*

$$\frac{1}{s_n} \sum_{i=1}^n (X_i - \mu_i) \xrightarrow{d} N(0, 1) \tag{1}$$

Give Definition 1 and Theorem 2, the following corollary follows naturally. The proof is deferred to Appendix C.

**Corollary 3** (IID after Sharding). *Assume that the updates from the clients follow Definition 1 where $k \ll \frac{n}{p}$. Besides, $\lim_{|H| \to \infty} \frac{1}{s_H^2} \sum_{i \in H} \mathbb{E}[(g_i - \mu_i)^2 \cdot \mathbb{1}\{|g_i - \mu_i| > \epsilon s_H\}] = 0$ where $s_H^2 = \sum_{i \in H} \sigma_{\phi(i)}^2$. Given the uniform randomness of sharding, we can view the distribution index $\phi(j)$ as drawn from some distribution $\Phi$ on $[k]$. Let $\bar{\mu} = \sum_{x \in [k]} \Phi(x = i)\mu_i$ and $\bar{\sigma}^2 = \sum_{x \in [k]} \Phi(x = i)\sigma_i^2$. Then,*

$$\frac{1}{|H|} \sum_{i \in H} g_i \xrightarrow{d} N(\bar{\mu}, \frac{\bar{\sigma}^2}{|H|})$$

.

**Empirical Validation.** To validate the claim empirically, we simulate heterogeneous data distribution by assigning MNIST samples with different labels to 25 clients. These clients are split into 5 shards. We plot the distributions of the updates before and after sharding as shown in Figure 1. Each line represents the weight distribution within one update. Figure 1a plots five updates from the same shard and Figure 1b plots the averaged updates from the five shards. It is obvious that after sharding the distributions are more densely and identically distributed as discussed above.

(a) Before sharding.  (b) After sharding.

Figure 1: Distribution w/ (w/o) sharding.

## 5 EVALUATION

In this section, we want to answer the following questions using empirical evaluation: (1) Does FilterL2 outperforms other aggregators when used alone? (2) Does $F^2$ED-LEARNING outperform other robust FL protocols augmented with sharded secure aggregation?

### 5.1 ATTACKS

To answer the above questions, we evaluated the robust estimators without attack and with several representative attacks. We focus on three of these attacks in the main text and defer one another attack to Appendix D.

The first and second attacks we used are the model poisoning attacks from Fang et al. (2019). The aim of the model poisoning attacks is to increase the error rate of the converged model even facing Byzantine-robust protocols. In these attacks, the malicious clients search for poisoning updates by solving an optimization problem. We employ two attacks proposed in their work targeting at Krum and Trimmed Mean. These two attacks are henceforth referred to as Krum attack (KA) and trimmed mean attack (TMA).

The third attack we considered is a backdoor attack from Bhagoji et al. (2019). The attack aims to insert a backdoor functionality while preserving high accuracy on the validation set. Similarly, the search for the attack gradient is formalized as an optimization problem and the authors tweak the objective function with some stealth metrics to make the attack gradient hard to detect. We refer to the attack as Model Poisoning Attack (MPA) in the rest of the section.

### 5.2 EXPERIMENTAL SETUP

We selected two datasets: MNIST (LeCun et al. (2010)) and FashionMNIST (Xiao et al. (2017)), and three other Byzantine-robust federated learning protocols to compare with: (1) Krum (Blanchard et al. (2017)); (2) Trimmed Mean (Yin et al. (2018)); and (3) Bulyan (Mhamdi et al. (2018)). Note that Bulyan acts like a wrapper around other robust estimators so in the evaluation we have two versions of Bulyan: Bulyan Krum and Bulyan Trimmed Mean. We ran all the protocols on the two datasets and present the attack performance under these protocols. Attack performance is measured differently according to the different attack targets. For KA and TMA, we use the model accuracy as the metric for characterizing attack performance. Higher model accuracy indicates stronger robustness. For MPA, we use the percentage of the remembered backdoors to represent the attack performance. The fewer backdoors remembered, the more robust the estimator is. In the IID setting, the data is shuffled and partitioned into all the clients, each receiving the same number of examples. In the non-IID setting, each client is assigned data with 3 labels. FilterL2 used in the evaluation is the accelerated version as discussed in Section 4.1.

### 5.3 EVALUATION RESULTS

**FilterL2 Performance.** To answer question (1), we evaluated 6 aggregators on MNIST and FashionMNIST as shown in Figure 2 and Figure 3. We ran the protocols with 20 clients with IID data distributions. 5 of the clients are malicious under attacks. Note that the number of the malicious clients actually exceeds the bound in Corollary 2 because some attacks only work with enough malicious clients.

Not surprisingly, FilterL2 achieves optimal performance among all 6 aggregators. Besides, FilterL2 is the only aggregator that consistently achieves good performance under all three attacks. The superiority of FilterL2 is owed to its quad-root estimation error. Due to the theoretically stronger robustness, it is extremely hard to design targeted attacks for FilterL2 like Krum or trimmed mean. In the non-IID setting, all the estimators cannot perform well since the IID assumption is broken.

$F^2$ED-LEARNING **Performance.** To answer question (2), we evaluated six aggregators with sharding on MNIST and FashionMNIST as shown in Figure 4 (homogeneous distribution) and Figure 5 (heterogeneous distribution). We ran the protocols with 100 clients, ten of which are malicious under attacks. The 100 clients were randomly split into 25 shards.

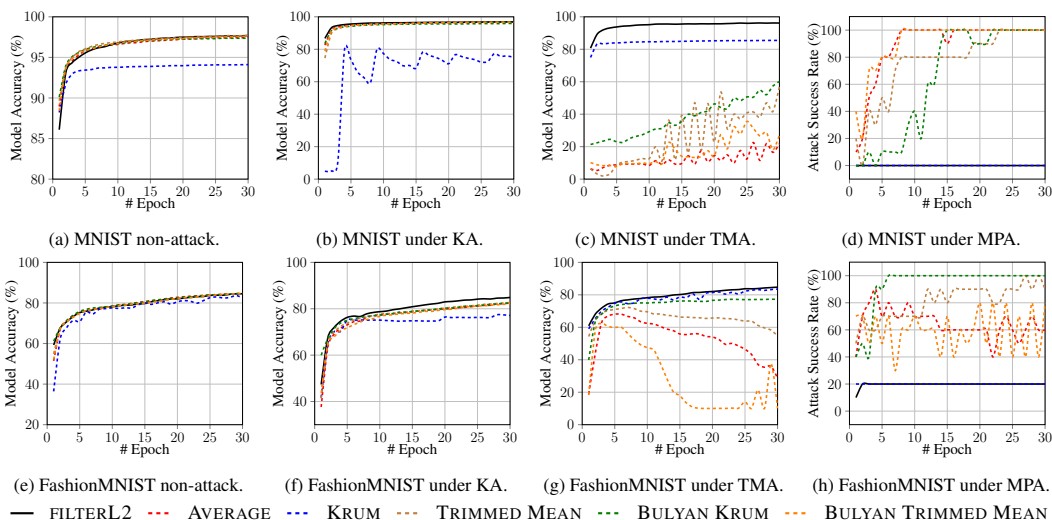

Figure 2: Attack performance under different Byzantine-robust estimators with IID data distribution.

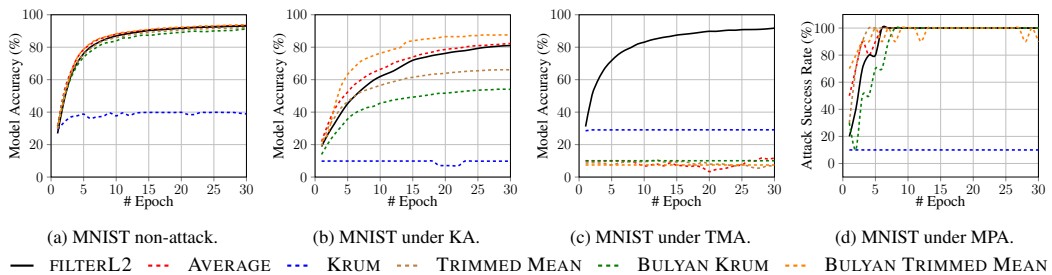

Figure 3: Attack performance under different estimators with non-IID data distribution.

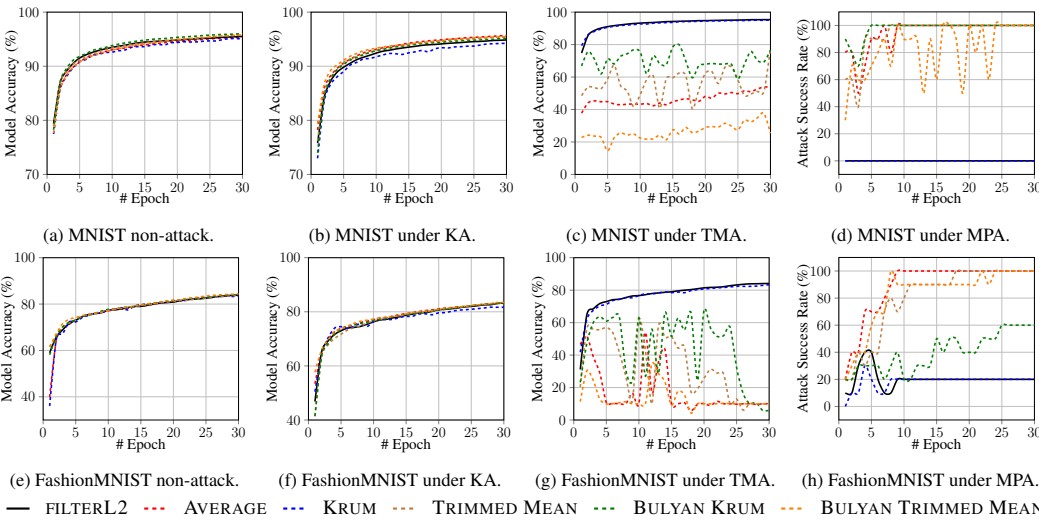

Figure 4: Attack performance under different estimators with sharded secure aggregation with IID data distribution.

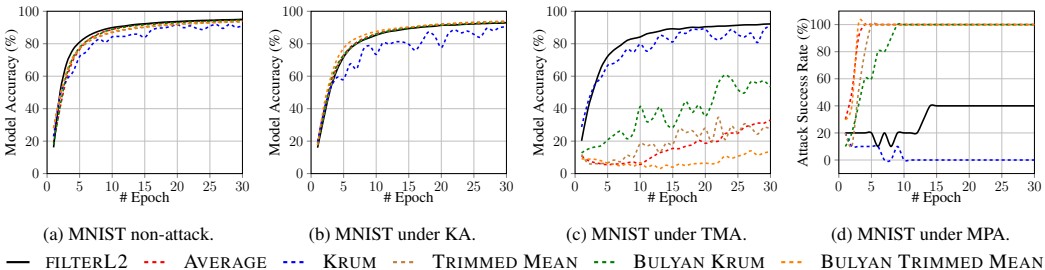

(a) MNIST non-attack. (b) MNIST under KA. (c) MNIST under TMA. (d) MNIST under MPA.

— FILTERL2 --- AVERAGE --- KRUM --- TRIMMED MEAN --- BULYAN KRUM --- BULYAN TRIMMED MEAN

Figure 5: Attack performance under different estimators with sharded secure aggregation with non-IID data distribution.

In both IID and non-IID settings, for the experiments without attack, with TMA or with MPA (Figure 4a,4c,4d,4e.4g,4h, 5a, 5c, 5d), F²ED-LEARNING still achieves optimal or close-to-optimal performance. An interesting phenomenon is that KA can be successfully defended by all aggregators when the clients are sharded (Figure 4b,4f). The reason is that KA is targeted at Krum without sharding and wants to maximize the probability that a malicious update is chosen by Krum.

Once integrated with sharding, Krum selects from the averaged updates from the shards, and thus the effect of the malicious update is diluted. This demonstrates that sharding itself can defend against some attacks by diluting the effect of malicious updates. Compared with the non-IID setting without sharding (Figure 3), all of the aggregators perform better under different attacks which benefit from the approximate IID distribution among shards.

**Influence of Shard Size.** As a new hyper-parameter: the number of shards $p$ (aka the shard size $\frac{n}{p}$), is introduced in F²ED-LEARNING, we empirically evaluate its influence as shown in Figure 6. The results in Figure 6 also clearly exhibit the trade-off between security and robustness guarantee via tuning $p$. When the number of shards equals the number of clients, the system is equivalent to FilterL2 without sharding and achieves optimal model accuracy. However, under such setting, the centralized server has access to each client's individual update and F²ED-LEARNING provides no further security guarantee than vanilla FL. When the number of shards converges to one, the system degrades to simple averaging with secure aggregation with the strongest security but the weakest robustness. When the number of shards lies between the two extremes, the model accuracy gradually changes under TMA as shown in Figure 6.

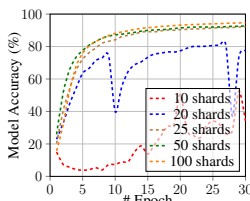

Figure 6: MNIST under TMA with different number of shards.

## 6 CONCLUSION & FUTURE DIRECTION

In this paper, we designed and developed F²ED-LEARNING, the first federated learning protocol defending against an honest but curious server and Byzantine malicious clients simultaneously. We propose to use FilterL2 to robustly aggregate the possibly contaminated updates and secure aggregation to protect the privacy of the clients. We reconcile the contradictory components with sharding. The evaluation results show that F²ED-LEARNING consistently achieves the optimal or close-to-optimal performance among five robust FL protocols. As far as we can see, F²ED-LEARNING addresses the two main privacy threats in FL systems simultaneously and shows the potential to further popularize FL in sensitive applications.

We also identify several unsolved challenges in F²ED-LEARNING which might motivate future works in FL with two-way protection. For instance, vanilla FilterL2 brings large overhead due to its high complexity. Although the accelerated FilterL2 partially addresses the issue, it sacrifices the asymptotic estimation error for the speedup. An interesting future direction is to integrate robust mean estimators with low complexity such as Cheng et al. (2019). However, Cheng et al. (2019)'s approach is rather complicated so designing low-complexity robust mean estimator with simple intuition is also an intriguing direction.

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

## A  PROOF OF COROLLARY 1

*Proof.* Corollary 1 is equivalent to the following lemma.

**Lemma 1** (Lemma 6.1 in Bonawitz et al. (2017)). *Given the number of the shards $p$, the parameter size $d$, the group size $q$, and the updates $\boldsymbol{g}^{H_i}$ where $\forall i \in [p], \boldsymbol{g}^{H_i} \in \mathbb{Z}_q^d$, we have*

$$\{\{\boldsymbol{u}_{ij} \xleftarrow{\$} \mathbb{Z}_q^d\}_{i<j}, \boldsymbol{u}_{ij} := -\boldsymbol{u}_{ji}, \forall i > j : \{\boldsymbol{g}^{H_i} + \sum_{j \in [p]/i} \boldsymbol{u}_{ij} \pmod q\}_{i \in [p]}\}$$

$$\equiv \{\{\boldsymbol{v}_i \xleftarrow{\$} \mathbb{Z}_q^d\}_{i \in [p]}\} \ s.t. \ \sum_{i \in [p]} \boldsymbol{v}_i = \sum_{i \in [p]} \boldsymbol{g}^{H_i} \pmod q : \{\boldsymbol{v}_i\}_{i \in [q]}\}$$

*where "$\equiv$" denotes that the distributions are identical.*

We prove the above lemma with induction on $n$.

**Base case:** When $n = 2$, the first elements in the RHS of two distributions are both uniformly random. The second element is the sum minus the first element. Thus the distributions are identical.

**Inductive Hypothesis:** We assume that when $n = k$, the two distributions are identical.

**Inductive Step:** If we ignore all the randomness from and for client $k + 1$, according to the inductive hypothesis, the distributions of the first $k$ clients are identical. Hence, after adding two independently uniformly random values, the distributions are still identical. The value for the $k+1^{th}$ client the the sum minus $k$ uniformly random values. Thus, the two distributions are still identical.

$\square$

## B    PROOF OF COROLLARY 2

*Proof.* In the following analysis, we assume that the updates from the shards follow an IID distribution. The reasonableness of the assumption is further discussed in Section 4.3.

The fraction of malicious shards is bounded by the worst case where each malicious client is exclusively assigned to different shards: $\epsilon' \leq \frac{\epsilon n}{p} \leq \frac{1}{12}$. Given the assumption above, we have satisfied all the requirements in Theorem 1. Hence, F$^2$ED-LEARNING provides a mean estimation with dimension-free error as long as $12\epsilon n < p$.

$\square$

## C    PROOF OF COROLLARY 3

*Proof.* By reorganizing Equation 1, we have $\frac{1}{|H|} \sum_{i \in H} g_i \xrightarrow{d} N(\frac{1}{|H|} \sum_{i \in H} \mu_{\phi(i)}, \frac{s_H^2}{|H|^2})$. Given the multinomial distribution $\Phi$,

- $\mathbb{E}[\frac{\sum_{i \in H} \mu_{\phi(i)}}{|H|}] = \sum_{i \in [k]} \Phi(x=i)\mu_i = \bar{\mu}, \mathbb{V}[\frac{\sum_{i \in H} \mu_{\phi(i)}}{|H|}] = \frac{\sum_{i \in [k]} \Phi(x=i)(\mu_i - \bar{\mu})^2}{|H|} = \frac{\sigma_{\mathbb{E}}^2}{|H|}$

- $\mathbb{E}[\frac{s_H^2}{|H|}] = \sum_{x \in [k]} \Phi(x=i)\sigma_i^2 = \bar{\sigma}^2, \mathbb{V}[\frac{s_H^2}{|H|}] = \frac{\sum_{x \in [k]} \Phi(x=i)(\sigma_i^2 - \bar{\sigma}^2)^2}{|H|} = \frac{\sigma_{\mathbb{V}}^2}{|H|}$

Thus, when $|H| \to \infty, \frac{\sum_{i \in H} \mu_{\phi(i)}}{|H|} \to \bar{\mu}, \frac{s_H^2}{|H|^2} \to \frac{\bar{\sigma}^2}{|H|}$.

$\square$

## D    MODEL REPLACEMENT ATTACK

In this section, we focus on the backdoor attack introduced in Bagdasaryan et al. (2020), namely model replacement attack. Similar to MPA, we use Attack Success Rate as the metric to measure the robustness of an estimator. The results are included in Figure 7.

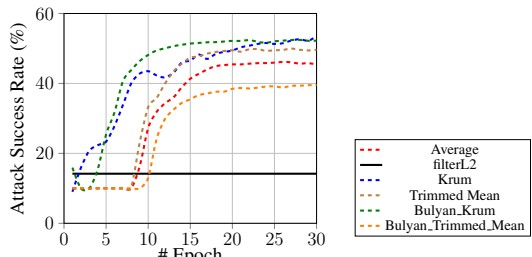

Figure 7: FashionMNIST under Model Replacement Attack Bagdasaryan et al. (2020).

