# OpenReview forum: "F^2ed-Learning: Good Fences Make Good Neighbors"
_ICLR.cc/2021/Conference — Reject_

### Official Review · AnonReviewer1 · 2020-10-21
**A protocol for protecting both clients and the server**

**Rating:** 5
**Confidence:** 2

**Review:**

The paper proposes a simple protocol to allow for both robust mean estimation and secure aggregation by sharding users, applying secure aggregation between shards and then doing robust mean estimation on the means returned by each shard.

Pros:

The proposed method is simple.

Experiments suggest the proposed method is more robust to various attacks than competitors.

Cons:

Generally the paper requires further editing as there are several places where the descriptions could be more clear (e.g. "achieves
optimal or sub-optimal performance").

In all Theorems it should be noted whether this result is by the authors or whether it is from Steinhardt (2018). Any Theorems which are quoted should be fully attributed, and any which are novel should be accompanied by formal proofs.

It is quite unclear what value the discussion of creating IID shards brings. Non-IID data between clients is a concern in federated learning, and it can cause issues such as diverging model parameters when each client takes multiple steps of GD locally. However this issue is not solved by sharding the users (as any averaging takes place after all local updates are made), nor is it clear if this solves any other issues. The authors should identify what value having these IID shards brings. Additionally it is unclear that the Lindeberg CLT is required here, as the shards draw directly from the mixture distribution induced by the clients $D_i$ distributions.

The paper would benefit from a much stronger explanation of the value of the IID-ness of the shards and more exploration of the value added by sharding.

---

> ### Author Response · Authors · 2020-11-20
> **Response to Reviewer 1**
>
> We thank the reviewer for the constructive comments. Please see below for our response.
>
> Q: Generally the paper requires further editing as there are several places where the descriptions could be more clear (e.g. "achieves optimal or sub-optimal performance").
>
> A: Thanks for pointing this out. We have extended the latest revision to clarify the confusions.
>
> Q: In all Theorems it should be noted whether this result is by the authors or whether it is from Steinhardt (2018). Any Theorems which are quoted should be fully attributed, and any which are novel should be accompanied by formal proofs.
>
> A: Thanks for pointing this out. We have added proof for Corollary 1 and 2 in Appendix A and B.
>
> Q: It is quite unclear what value the discussion of creating IID shards brings. Non-IID data between clients is a concern in federated learning, and it can cause issues such as diverging model parameters when each client takes multiple steps of GD locally. However this issue is not solved by sharding the users (as any averaging takes place after all local updates are made), nor is it clear if this solves any other issues. The authors should identify what value having these IID shards brings. Additionally it is unclear that the Lindeberg CLT is required here, as the shards draw directly from the mixture distribution induced by the clients Di distributions. The paper would benefit from a much stronger explanation of the value of the IID-ness of the shards and more exploration of the value added by sharding.
>
> A: We apologize for the confusion caused by our insufficient elaboration. The value of sharding is two-fold: (1) FilterL2 is proved to be robust on IID inputs. However, in FL, the updates from individual clients are known to be non-IID. Hence, we create the IID shards to bridge the gap. The Lindeberg CLT is used to argue why the updates after aggregation within shards approximately follow a proper IID distribution (among shards) with bounded variance. We have added Corollary 3 in the latest revision to make our claim more clear. (2) Sharding is also required to reconcile secure aggregation and robust aggregation. As we mentioned in the paper (section 1), secure aggregation hides individual updates while robust aggregation requires those information. Therefore by tweaking the shard size, we can conveniently trade off the security guarantee and the robustness guarantee. We have reorganized and rewritten Section 4.3 to clarify the value of IID sharding.

---

> > ### Comment · AnonReviewer1 · 2020-11-23
> > **Corollary 3**
> >
> > Corollary 3 seems suggest that after sharding the resulting r.v. will be IID. However if you consider the r.v. being put into each shard as non-IID, then the r.v. after sharding is still non-IID (even in the limit), since each shard would converge to a normal with different means and variances.
> >
> > Given the importance of this IID/non-IID discussion the paper would benefit from having a well specified description of the distributions being discussed.

---

> > > ### Author Response · Authors · 2020-11-23
> > > **Some clarification about Corollary 3**
> > >
> > > Thanks for the quick response! We clarify Corollary 3 from the following two perspectives.
> > >
> > > **Reasonability of Corollary 3.** We totally agree that if the r.v. in each shard is completely IID, then the r.v. after sharding is still non-IID. However, given our model for heterogeneous distribution as presented in Definition 1, the r.v. is drawn from *a finite family of different distributions*. Hence, following the CLT, an averaged update from a shard asymptotically follows a normal distribution if the size of the distribution family is several times smaller than the shard size. The expectation and scale of the normal distribution again follow two meta-distributions with bounded variance. Thus according to Chebyshev's inequality, w.h.p. the updates from shards *asymptotically approximately follow* an IID distribution. We do have the concern about the influence of the approximation. The results in Figure 1 empirically support the claim. The new empirical evaluation in Figure 3 and 5 also partially confirms that F2ED-LEARNING does work even under heterogeneous data distribution.
> > >
> > > **Reasonability of Definition 1.** The above discussion is completely based on Definition 1. A natural question is "why is the Definition 1 reasonable?". Actually, Definition 1 is abstracted from the common agreement that "samples within the same classification class follow an IID distribution". Thus, Definition 1 applies to many practical scenarios. For instance, if a health app wants to train a model for diabetes prediction using FL, then the samples from each client will be drawn from the distribution w/ or w/o diabetes and thus follow Definition 1. Following the same logic, Definition 1 can even accommodate more complex scenarios, where samples from each client is a mixture of difference classes but the number of samples is limited, by taking a combinatory argument.
> > >
> > > The newest response to AnonReviewer2 includes an example which might help illustrate our claim. We have added a constraint that $k\ll \frac{n}{p}$ in Corollary 3 in the latest revision and will further clarify Corollary 3 and Definition 1 in the next revision. We would appreciate it if the reviewers have any follow-up comments.

---

> > > > ### Comment · AnonReviewer1 · 2020-11-23
> > > > **Constraint on k**
> > > >
> > > > Adding this new constraint on k does deal with this issue, but does it not also effectively eliminate non-IIDness (of clients). In such a situation is sharding still useful/necessary?

---

> > > > > ### Author Response · Authors · 2020-11-23
> > > > > **Necessity of sharding**
> > > > >
> > > > > Thanks for the follow-up question! Sharding is necessary due to the following reasons.
> > > > >
> > > > > First, Definition 1 captures many practical scenarios where traditional IID hypothesis might not fit. For instance, updates from each client follow either $N(0, 1)$ with probability $0.1$ or $N(100, 1)$ with probability $0.9$. If we directly apply FilterL2 under IID assumption, it is likely that most points from $N(0,1)$ will be filtered out and the mean estimation is $100$. If we create shards with size $100$, then the updates from shards are expected to follow $N(90, 0.82)$ and we can report a relatively accurate mean estimation. This ideal example captures important features of many real-world FL tasks. For example, for a classification task, if each client can only hold data with the same label, then the clients with the same label approximately follow IID distribution and the number of distributions is equal to the number of classes which is likely to be much smaller than the shard size.
> > > > >
> > > > > Besides, we would like to emphasize that the main purpose of sharding is to reconcile security and robustness. The discussion about IID-ness via sharding is more of a bonus introduced by the sharding technique. With such bonus, we can deal with some non-IID cases in FL which cannot be handled under IID hypothesis. We will tune down our claim a little bit in the paper correspondingly.
> > > > >
> > > > > We would appreciate it if the reviewer has further comments.

---

### Official Review · AnonReviewer3 · 2020-10-25
**Interesting idea, some concerns about the theory and experimental setup**

**Rating:** 5
**Confidence:** 4

**Review:**

##########################################################################

Paper summary:

The paper considers robustness to poisoning and backdoor attacks in the context of federated learning. It proposes a defence based  on splitting the clients into shards, averaging their updates via secure aggregation and then using a robust mean estimation on top to ensure robustness. The authors point out that controlling the number of shards is a way to trade-off privacy vs robustness, thus potentially dealing with both malicious clients and an honest, but curious server. The paper provides some theoretical justification for the algorithm, as well as an experimental evaluation where its performance is tested against multiple attacks and compared to other existing methods.

##########################################################################

Pros:

- The paper studies an important problem. Federated learning is an increasingly popular way of training large-scale models. However, the privacy that it offers comes with the cost of vulnerability to training time attacks. Hence, the problem of guaranteeing robustness while preserving privacy is certainly important.
- I find the idea of splitting clients into groups to trade-off privacy vs robustness very interesting. To my awareness the idea is novel in the context of robust federated learning.
- Overall, the paper is well-written and it tries to justify the proposed method both theoretically and empirically.


##########################################################################

Cons:

1. While the papers aims to address the problem of robust federated learning, most of its theoretical analysis, as well as the optimization procedure that is described, is actually tailored to a classic (i.i.d.) distributed learning setting.

- The analysis is Section 4.1 and 4.2 assumes that the data of the good clients is i.i.d.. In section 4.3 an informal argument is made as to why this can be safely ignored because of the averaging inside the shards. However, I find the argument unconvincing for two reasons. First, the CLT argument is tailored to scalar random variables. Applying it to each dimension independently does not seem trivial to me and, if at all possible, might involve an unfortunate dependence in the dimension of the problem. Secondly, even on the intuitive level the resulting distributions will be close to identical only when all clients are good. If some of them can behave arbitrarily, they can shift the mean of their shard, thus breaking the i.i.d. property.

- Perhaps a smaller issue is that the plain optimization method considered in the paper (which is modified via the robust aggregation procedure later on) is SGD. In contrast, many federated learning algorithms, for example, FED-AVG, are based on averaging model parameters and in addition train a model for every client individually. It is unclear whether the analysis in this paper would transfer there as well.

- The baselines considered in the experimental section are also taken from papers that study the i.i.d. version of the problem. Since little information is given about how the training data is distributed across clients, it's hard to know if the comparison to the i.i.d. baselines is fair and also if the experimental setup corresponds to i.i.d. or non-i.i.d. data.

2. Overall, the theoretical analysis in the paper appears insufficient and is not backed up by any proofs.

- Some of the presented theorems and corollaries seem to come from prior work, while others seem to be novel. I am assuming that Theorem 1 is from Steinhardt (2018), while Theorem 2 and 3 are novel. I think this should be made more clear.
- Theorem 2 and 3 are just stated without any proof. I was also unable to find proofs in the supplementary material. This, together with the lack of intuitive explanation about why these results should hold, makes it impossible to judge the validity and novelty of these results.
- Theorem 3 states that for small enough number of Byzantine workers, a dimension independent error can be obtained. To me this sounds rather vague. Does this mean that the mean of the true gradients can be estimated to a dimension-independent accuracy at each time step? Or does it mean that at the end the algorithm converges to an epsilon-stationary point, with a number of steps that features no dependence on the dimension of the problem? How would this compare to results in previous work on Byzantine robustness?

3. While experiments are provided on two datasets and against a large amount of attacks, some of the comparisons seem unfair to me.

- In particular, all attacks apart from the backdoor one are tailored against some of the baselines. Naturally, the corresponding baselines compare badly against the attacks tailored to them. At the same time, Krum performs quite well against trimmed mean attacks and similarly the trimmed mean works well against the Krum attack. It therefore would be more fair to also create an attack specifically towards the algorithm proposed in the paper and check the performance under this attack. Alternatively, defence-independent attacks can be used.
- It would be nice to include more details about how the data was split among workers, so that i.i.d. and non-i.i.d. situations can be spaced out.
- In the first set of experiments, what is the value of p? It would be interesting to see how the proposed algorithm performs for various values of p, both in terms of robustness and in terms of some notion of privacy.

##########################################################################

Review summary:

I find the high-level idea and the approach taken in the paper quite interesting. However, due to a few concerns about the theoretical justification and the experimental setup I do not recommend acceptance. I believe that a more detailed theoretical analysis and a wider set of experiments are needed to strengthen the submission and make it easier to compare it to prior work.

---

> ### Author Response · Authors · 2020-11-20
> **Response to Reviewer 3 (Part 1)**
>
> We thank the reviewer for the constructive comments. Please see below for our response.
>
> Q: The analysis is Section 4.1 and 4.2 assumes that the data of the good clients is i.i.d.. In section 4.3 an informal argument is made as to why this can be safely ignored because of the averaging inside the shards. However, I find the argument unconvincing for two reasons. First, the CLT argument is tailored to scalar random variables. Applying it to each dimension independently does not seem trivial to me and, if at all possible, might involve an unfortunate dependence in the dimension of the problem. Secondly, even on the intuitive level the resulting distributions will be close to identical only when all clients are good. If some of them can behave arbitrarily, they can shift the mean of their shard, thus breaking the i.i.d. property.
>
> A: Thanks for the detailed comment. (1) The effect of CLT on the estimation error is a very interesting question. We deem this as an open problem and will try to solve it in the future work. On the other hand, we suppose this challenge universally exists in both traditional distributed learning and federated learning as no real-world updates are exactly IID distributed. Thus we do not regard the error introduced by the CLT as a unique defect of F2ED-LEARNING. Instead, it is a universally existing issue when you apply an IID-based algorithm to real-world tasks.  (2) We are not afraid that malicious clients break the IID property. Indeed, we only require the shards without malicious clients to follow an IID distribution. Actually, what the reviewer described is a typical attack scenario and can be effectively defended by FilterL2.
>
> Q: Perhaps a smaller issue is that the plain optimization method considered in the paper (which is modified via the robust aggregation procedure later on) is SGD. In contrast, many federated learning algorithms, for example, FED-AVG, are based on averaging model parameters and in addition train a model for every client individually. It is unclear whether the analysis in this paper would transfer there as well.
>
> A: Thanks for the comment. Personalized FL is of orthogonal interest to F2ED-LEARNING. Besides, we do not see any challenge preventing F2ED-LEARNING from being used in personalized FL to train the global model.
>
> Q: The baselines considered in the experimental section are also taken from papers that study the i.i.d. version of the problem. Since little information is given about how the training data is distributed across clients, it's hard to know if the comparison to the i.i.d. baselines is fair and also if the experimental setup corresponds to i.i.d. or non-i.i.d. data.
>
> A: Thanks for the helpful comments! The data is IID distributed across clients in all the evaluation to make the comparison fair. We have added several evaluations with non IID data distribution in the latest revision. We are running more evaluation with heterogeneous updates and will report the results as soon as we get them.
>
> Q: Overall, the theoretical analysis in the paper appears insufficient and is not backed up by any proofs. Some of the presented theorems and corollaries seem to come from prior work, while others seem to be novel. I am assuming that Theorem 1 is from Steinhardt (2018), while Theorem 2 (Corollary A in the latest revision) and 3 (Corollary B in the latest revision) are novel. I think this should be made more clear. Theorem 2 (Corollary A in the latest revision) and 3 (Corollary B in the latest revision) are just stated without any proof. I was also unable to find proofs in the supplementary material. This, together with the lack of intuitive explanation about why these results should hold, makes it impossible to judge the validity and novelty of these results.
>
> A: Thanks for pointing this out. We have added proof for Corollary 1 and 2 in Appendix A and B. Besides, we add more theoretical analysis (Corollary 3) in Section 4.3 to clarify some confusions about the value of IID sharding. Moreover, we have made the attribute of each theorem or corollary more clear in the latest revision.

---

> > ### Author Response · Authors · 2020-11-20
> > **Response to Reviewer 3 (Part 2)**
> >
> >
> > Q: Theorem 3 states that for small enough number of Byzantine workers, a dimension independent error can be obtained. To me this sounds rather vague. Does this mean that the mean of the true gradients can be estimated to a dimension-independent accuracy at each time step? Or does it mean that at the end the algorithm converges to an epsilon-stationary point, with a number of steps that features no dependence on the dimension of the problem? How would this compare to results in previous work on Byzantine robustness?
> >
> > A: It means that “the mean of the true gradients can be estimated to a dimension-independent accuracy at each time step”. Once we have this per-step error bound, according to some standard convergence result such as Corollary 2.2 in [1], we can easily relate the per-round error rate with the global convergence rate and obtain a better convergence rate. A thorough comparison between F2ED-LEARNING and other robust aggregators is made in Section 2. In a word, F2ED-LEARNING is the only aggregator with dimension-free per-round error against general Byzantine attacks. Given the parameter dimension is huge for deep network (e.g. $10^5\sim 10^7$), the saving is huge.
> >
> > Q: While experiments are provided on two datasets and against a large amount of attacks, some of the comparisons seem unfair to me. In particular, all attacks apart from the backdoor one are tailored against some of the baselines. Naturally, the corresponding baselines compare badly against the attacks tailored to them. At the same time, Krum performs quite well against trimmed mean attacks and similarly the trimmed mean works well against the Krum attack. It therefore would be more fair to also create an attack specifically towards the algorithm proposed in the paper and check the performance under this attack. Alternatively, defence-independent attacks can be used.
> >
> > A: Thanks for the constructive comment! We have the same concern with the reviewer. However, we finally chose to set the experiment up as it is due to the following two reasons. (1) Designing attack for F2ED-LEARNING is **not trivial**. We attempted to design targeted attacks on F2ED-LEARNING following the methodology in [2]. However, it is unclear how to solve the optimization problem (1) in [2] if we bring FilterL2 in it. Besides, even well-designed attacks cannot break the dimension-free guarantee since it is proven. Thus, we cannot foresee any effective attack running in polynomial time and we deem designing attacks on estimators with dimension-free error an interesting open problem. (2) We have tested many general-purpose attacks such as [3] and [4]. However, these attacks do not work on every robust aggregator. The authors of [4] also acknowledge in the paper that their attack cannot consistently succeed even under weak robust aggregators such as Krum and BULYAN. Thus, it is meaningless to evaluate the defence on these attacks.
> >
> > Q: It would be nice to include more details about how the data was split among workers, so that i.i.d. and non-i.i.d. situations can be spaced out.
> >
> > A: In current (IID) experiments, the data is shuffled and partitioned into n clients, each receiving the same number of examples. We have also added some non-IID experiments in the latest revision and are planning to add more. In non-IID evaluations, we divide the data by label, assign each client with a share of the data, and then randomly partition the clients into shards. Please refer to Section 5.1 for more details.
> >
> > Q: In the first set of experiments, what is the value of $p$? It would be interesting to see how the proposed algorithm performs for various values of p, both in terms of robustness and in terms of some notion of privacy.
> >
> > A: The first set of experiments is designed to test FilterL2 without sharding so there is no hyper-parameter $p$. We have started the experiments on various sharding sizes and will report the results as soon as we get them.
> >
> > [1] Ghadimi, Saeed, and Guanghui Lan. "Stochastic first-and zeroth-order methods for nonconvex stochastic programming." SIAM Journal on Optimization 23.4 (2013): 2341-2368.
> > [2] Fang, Minghong, et al. "Local model poisoning attacks to Byzantine-robust federated learning." 29th {USENIX} Security Symposium ({USENIX} Security 20). 2020.
> > [3] Bagdasaryan, Eugene, et al. "How to backdoor federated learning." International Conference on Artificial Intelligence and Statistics. PMLR, 2020.
> > [4] Xie, Chulin, et al. "DBA: Distributed Backdoor Attacks against Federated Learning." International Conference on Learning Representations. 2019.

---

> > > ### Comment · AnonReviewer3 · 2020-11-22
> > > **The theoretical analysis is heavily improved, I'm still unconvinced about the experiments though.**
> > >
> > > Thank you for the updates and the detailed response. The extended discussion around the theoretical results and the CLT argument have significantly improved the exposition of the paper. I have updated my rating accordingly.
> > >
> > > I am still uncomfortable with the comparisons made in the experimental section though. I understand that designing an attack against F^2ed-learning is challenging, which is in fact another argument in favour of the algorithm. However, I do not see why results against defence-independent attacks are not reported - even if they do not always work well against the baselines, it's important to see if they work against the proposed algorithm. The current poisoning attacks work well against the defence that they target, but not against the other baselines. In this sense, I find it hard to compare the effectiveness of F^2ed-learning to the effectiveness of the baselines based on the current set of experiments.
> > >
> > > I do also think that experiments with various sharding sizes are important to report, especially in the context of the privacy-robustness trade-off discussed in the theoretical section, so I'm looking forward to your follow-up.

---

> > > > ### Author Response · Authors · 2020-11-23
> > > > **Some initial results for the required evaluation**
> > > >
> > > > We really appreciate the quick feedback! We are planning to improve the evaluation section as suggested and we would like to report some initial results we have got before the discussion period ends.
> > > >
> > > > **More attacks.** After the discussion, we do agree with the reviewer that the results for defense-independent attacks should be reported. We have started experiments with Model Replacement Attack [1] and DBA Attack [2]. Some initial results are reported in Appendix D in the latest revision. The initial results show that F2ED-LEARNING still achieves optimal performance although the attack is not very effective on other aggregators (even simple averaging) as well.
> > > >
> > > > **Influence of Sharding Size.** We have started experiments with different sharding sizes and some initial results are reported at the end of Section 5.3 in the latest version. We can see a clear trade-off between security and robustness as the number of shards change from 1 to the number of clients and the model accuracy gradually changes between the two extremes.
> > > >
> > > > [1] Bagdasaryan, Eugene, et al. "How to backdoor federated learning." International Conference on Artificial Intelligence and Statistics. PMLR, 2020.
> > > > [2] Xie, Chulin, et al. "DBA: Distributed Backdoor Attacks against Federated Learning." International Conference on Learning Representations. 2019.

---

### Official Review · AnonReviewer4 · 2020-10-28

**Rating:** 6
**Confidence:** 3

**Review:**

Summary: The authors consider federated learning setting and how to defend the overall learning task against malicious clients and a semi-honest centralized server. Though there are known ways to prevent attacks, they suffer from a large error in the estimator and also do not preserve privacy of updates since the server sees them in the clear in order to adjust for error. This paper proposes a sharding technique and use of the estimator method whose error does not depend on the number of dimensions as previous work.

Novelty:
1. the sharding approach allows the proposed method to use masking-like techniques to avoid the server seeing values of individual clients. That is, the server only sees aggregates in the shard
2. the paper proposes a different tradeoff in terms of the error of the estimate compared to mechanisms in related work. In particular, the error depends on the proportion of “malicious” clients.

Overall it is a very nicely written and presented paper.
I would suggest that the authors expand evaluation section to compare performance of the methods besides accuracy. That is how many rounds each algorithm takes, total communication cost and computation time of each approach.

Also I was not clear if one needs to make assumptions on the knowledge of the proportion of malicious clients in order to carry out the algorithm (Alg 2). Would that be known or there would be a known upper bound?

Please state if there is an assumption on non collusion between the server and the clients.

---

> ### Author Response · Authors · 2020-11-20
> **Response to Reviewer 4**
>
> We thank the reviewer for the positive feedback. Please see below for our response.
>
> Q: Also I was not clear if one needs to make assumptions on the knowledge of the proportion of malicious clients in order to carry out the algorithm (Alg 2). Would that be known or there would be a known upper bound?
>
> A: Yes we need an assumption on the proportion of malicious clients for Corollary 2 to hold. Specifically, we need the number of malicious clients to be less than $\frac{1}{12}$ of the number of shards. The constant $\frac{1}{12}$ is inherited from Theorem 1 which is required for some constant analysis to go through. We have made the assumption more clear in Corollary 2 in the latest revision.
>
>
> Q: Please state if there is an assumption on non collusion between the server and the clients.
>
> A: Yes we do require the non collusion between the server and the clients. We have made the non-collusion assumption explicit in the latest revision.

---

### Official Review · AnonReviewer2 · 2020-10-28
**Nice algorithm but missing some justifications**

**Rating:** 6
**Confidence:** 3

**Review:**

**Paper summary**

The paper claims to be the first paper that simultaneously handles Byzantine threats while ensuring privacy in a federated learning setup. One of their main claims is that this is the first algorithm that provides dimension independent robustness guarantees against byzantine threats (I have some concerns regarding this claim). The algorithm first divides all the machines into shards. Within each shard there is secure aggregation. Finally, the outputs of each shard is robustly aggregated such that the error isn't dimension dependent.

**Strengths**
1. The problem of handling privacy and tolerance to Byzantine adversaries (including data poisoning adversaries) is indeed very important and relevant to federated learning.
2. The paper provides dimension independent guarantees against Byzantine adversaries, which is of critical importance since modern models can have huge dimensionality.
3. The experiments show that the algorithm is better than other Byzantine resilient algorithms.

I like the clarification about the assumptions of Bulyan done in Section 2. This will make comparisons between related works easier.

**Concerns**
1. The proof of Theorem 2 seems to be missing.
2. I can see that Theorem 3 might be a direct corollary of Proposition 4.1 from (Steinhardt, 2018), but it would still be good to prove Theorem 3 for completeness.
3. I am not able to see how sharding helps non-iid data become iid. Corollary 1 (CLT) says that $\frac{1}{s_n}\sum_{i=1}^n (X_i - \mu_i) \stackrel{d}{\to} \mathcal{N}(0,1) $. However, what is needed for sharding to make data become iid would be something like $\frac{1}{n}\sum_{i=1}^n X_i \stackrel{d}{\to} \mathcal{N}(0,1)$. Note that there are two differences: The major difference is that we have $X_i$ instead of $X_i-\mu_i$. The second difference is we have $n$ in the denominator instead of $s_n$. Below, I give a concrete example how the CLT is not applicable for sharding.

Assume that we have $2n$ machines. We divided them into two shards of $n$ machines each (machines $1$ to $n$ go into shard $1$, and machines $n+1$ to $2n$ go into shard $2$). Now, what the paper claims would be that the mean of the two shards should converge to the same distribution, regardless of the distribution of the individual machine gradients. Now, assume that for some particular iteration, the gradient of the machines have the following distribution: Machines $1$ to $n$ have (iid) gradients equal to $1$ with probability $1$. Machines $n+1$ to $2n$ have (iid) gradients equal to $-1$ with probability $1$. Clearly, the mean of the first shard is the constant random variable $1$, whereas the mean of the second shard is the constant random variable $-1$. These two random variables don't have the same distribution. Thus, the distribution of the means of the shards is not identical and hence not iid.

4. If concern 3 is valid, then the algorithm would not work on heterogenous data. Further even if concern 3 is invalid the CLT only gives asymptotic result. A key point of this paper is dimension independent resilience. However, if the convergence given by the CLT is very slow as the dimensions increase, then the dimension independent resilience may no longer be valid.

For these reasons, I think the paper needs some more justification for its theoretical claims.


**Suggestions**

I think there are some papers that use the dimension-independent robust mean estimation techniques for non-FL learning. For example (Yin et al., 2019) (this is different from the one cited in your paper). Also, Alistarh et al.(2018) give dimension independent guarantees. It would be good to talk about these in related works.

**Score justification**

As mentioned in the Concerns section, I think the paper might be missing some justifications for its theoretical claims.

**Post Author Feedback Comments**

The authors have tried to address my main concern by adding an assumption on the distributions of gradients. Essentially they have assumed that the gradient distributions are sampled from a small, finite set of distributions. I don't know how realistic this assumption is, because each client can potentially have a different distribution. However, in practice, this may be approximately true. A better assumption would have been based on probability distances (TV distance, Wasserstein distance, etc.).

I have increased my score to 6.

**References**


Yin, D., Chen, Y., Kannan, R. and Bartlett, P., 2019, May. Defending against saddle point attack in Byzantine-robust distributed learning. In International Conference on Machine Learning (pp. 7074-7084). PMLR.
Alistarh, D., Allen-Zhu, Z. and Li, J., 2018. Byzantine stochastic gradient descent. In Advances in Neural Information Processing Systems (pp. 4613-4623).

---

> ### Author Response · Authors · 2020-11-20
> **Response to Reviewer 2**
>
> We thank the reviewer for the constructive comments. Please see below for our response.
>
> Q: The proof of Theorem 2 (Corollary 1 in the latest revision) seems to be missing. I can see that Theorem 3 (Corollary 2 in the latest revision) might be a direct corollary of Proposition 4.1 from (Steinhardt, 2018), but it would still be good to prove Theorem 3 for completeness.
>
> A: Thanks for pointing this out. We have included the proofs in Appendix A and B.
>
> Q: I am not able to see how sharding helps non-iid data become iid. Corollary 1 (CLT) says that $\frac{1}{s_n}\sum_{i=1}^n(X_i−\mu_i)\overset{d}{\rightarrow}N(0,1)$. However, what is needed for sharding to make data become iid would be something like $\frac{1}{n}\sum_{i=1}^nX_i\overset{d}{\rightarrow}N(0,1)$. Note that there are two differences: The major difference is that we have $X_i$ instead of $X_i−\mu_i$. The second difference is we have n in the denominator instead of $s_n$.
>
> A: We apologize for the confusion caused by our insufficient elaboration. Since $s_n$ and $\mu_i$ are constants, they can be easily moved to the RHS of the formula in Theorem 2. Then we can show that the updates from shards approximately follow an IID distribution with bounded variance. We have added Corollary 3 in the latest revision to clarify the issue.
>
> Q: If concern 3 is valid, then the algorithm would not work on heterogeneous data. Further even if concern 3 is invalid the CLT only gives asymptotic result. A key point of this paper is dimension independent resilience. However, if the convergence given by the CLT is very slow as the dimensions increase, then the dimension independent resilience may no longer be valid.
>
> A: Thanks for the comment. First, as we clarified in the above answer, we believe that our algorithm still works on heterogeneous data. Second, the shard updates converge to IID with the increase of the number of clients in each shard, not the dimension of the parameters. Thus, we do not see it changing the dimension-independent claim. We agree that sharding only provides approximate IID guarantee. However, we deem this as good enough since it is almost impossible to guarantee true IID in practice and our evaluation has confirmed the superiority of F2ED-LEARNING over other robust aggregators from the empirical perspective.
>
> Q: I think there are some papers that use the dimension-independent robust mean estimation techniques for non-FL learning. For example (Yin et al., 2019) (this is different from the one cited in your paper). Also, Alistarh et al.(2018) give dimension independent guarantees. It would be good to talk about these in related works.
>
> A: Thanks for pointing out the two papers.
>
> (1) [1] studies a pretty specific attack, the saddle point attack, instead of general Byzantine attacks. Besides, their error bound still suffers from a $\sqrt{d}$ factor where $d$ is the dimension of parameters as stated in Theorem 5 in [1].
>
> (2) [2] shares the same loophole as we pointed out in BULYAN. In Assumption 2.2 in [2], they assume the $l_2$ norm between the oracle output and the true gradient is bounded by $\mathcal{V}$. We point out that the assumption is much weaker than ours and actually hides a $\sqrt{d}$ factor implicitly (refer to Section 2 in our paper for more details). Thus their Byzantine robustness guarantee is still dimension-dependent. Besides, in the paper, the authors only discuss the convex optimization scenario. It is not clear whether the approach generalizes to non-convex optimization problems.
> Henceforth we believe that F2ED-LEARNING is still the first FL system with real dimension-free error. We have discussed these two papers in the related work Section in the latest revision.
>
> [1] Yin, D., Chen, Y., Kannan, R. and Bartlett, P., 2019, May. Defending against saddle point attack in Byzantine-robust distributed learning. In International Conference on Machine Learning (pp. 7074-7084). PMLR.
> [2] Alistarh, D., Allen-Zhu, Z. and Li, J., 2018. Byzantine stochastic gradient descent. In Advances in Neural Information Processing Systems (pp. 4613-4623).

---

> > ### Comment · AnonReviewer2 · 2020-11-23
> > **I still don't think the shards would be IID**
> >
> > If you move $\mu_i$ to the other side of the equation, then again each shard will have a different distribution because their means would be different (they would be $\mu_i$). Consider this case: We are in $n$ dimensions with $n$ workers. Worker $i$ has mean $e_i$ which is the vector of all zeros except a one at the $i$-th coordinate. Hence $e_1=[1\ 0\ 0\ 0\ \dots\ ], e_2=[0\ 1\ 0\ 0\ \dots\ ]$ and so one. Now, no matter how these machines are sharded, the shards will have different means and hence different distributions.
> >
> > I think AnonReviewer1 also has a similar concern.

---

> > > ### Author Response · Authors · 2020-11-23
> > > **Some clarification about Corollary 3 and Definition 1**
> > >
> > > Thanks for the quick feedback! We have further clarified Corollary 3 and Definition 1 in the newest response to AnonReviewer 1, which might be helpful to address the reviewer's concern. Here we further explain the corollary and definition based on the example the reviewer gave. Note that we add a new constraint in Corollary 3 $k\ll \frac{n}{p}$ to help understand.
> > >
> > > Given Definition 1 with the constraint $k\ll \frac{n}{p}$, the example does not hold because it is impossible that each client holds samples from a different distribution. An ideal and simplified scenario is that client $i$ holds samples $e_{i\mod k}$. Then with the randomness in sharding, the updates from the shards **asymptotically and approximately** follow an IID distribution. It is hard to measure the influence of the approximation theoretically but the results in Figure 1, 3, 5 partially confirm the claim empirically.
> > >
> > > We will try to give better elaboration in the next version and would appreciate it if the reviewer has further follow-up questions.

---

> > ### Comment · AnonReviewer2 · 2020-11-23
> > **Convergence rate of CLT wrt dimensions**
> >
> > Let me expand more on my point about the convergence rate of CLT with respect to dimensions. The CLT just says that appropriately  shifted and scaled empirical mean converges to the standard normal. It doesn't say anything about the rate of convergence. Now, there are theorems that give the convergence rate, for example the Berry-Esseen theorem. I am not familiar with the multi-dimensional version of the Berry-Esseen theorem, but a quick Google search shows that the convergence rate given by the multi-dimensional version has a $d^{1/4}$ dependence on the convergence rate. I may have understood the multi-dimensional version of the Berry-Esseen theorem wrongly or there may be a better convergence rate in multiple dimensions, but my point is that I do not think we can simply say that the convergence rate doesn't depend on the dimensions.
> >
> > Hence, I believe that there should be a discussion about this.

---

> > > ### Author Response · Authors · 2020-11-23
> > > **Thanks for the further explanation! We will add discussion in the next version.**
> > >
> > > Thanks for the further explanation of your concern. We now better understand the concern and do agree that the concern is well-established. We will look into the issue and add a remark to discuss the influence of CLT on the dimension-free error guarantee.

---

### Decision · Program_Chairs · 2021-01-07
**Final Decision**

**Decision:**

Reject

**Comment:**

The paper considers federated learning in the presence of malicious clients and a semi-honest centralized server. The authors provide a novel secure aggregation technique (i.e. split the clients into shards, and securely aggregate each shard’s updates, and the estimating things based on the updates from different shards) to protect clients from the server. Furthermore, an important property of the proposed protocol is that the estimation error is (provably) dimension-free against Byzantine malicious clients. The paper is well-written.

The reviewers had a number of concerns many of which were addressed during the rebuttal phase. There was also another round of discussion after the rebuttal phase. Overall, the reviewers felt that there are still some issues that need to be resolved (see the updated reviews--the main issues are: (i) the assumption of non-collusion between the server and the clients, (ii) assumptions and analysis of the non-iid case, and (iii)  comparing to attacks that are specifically targeted against the baselines). I believe that once these issues are addressed, the paper will provide an important contribution to the area of federated learning.